# Longitudinal association between psychological distress and mask-wearing post COVID-19 among psychiatric outpatients in Japan

Kazuhiro Suzuki[1,2,3]*, Yuka Mizuno[1,3,4], Yusuke Arai[1,2], Keitaro Miyamura[5], Daimei Sasayama[2], Shinsuke Washizuka[1,2]

**1** Department of Community Mental Health, Shinshu University School of Medicine, Matsumoto, Japan, **2** Department of Psychiatry, Shinshu University School of Medicine, Matsumoto, Japan, **3** Department of Psychiatry, Maruko Central Hospital, Ueda, Japan, **4** Department of Neuropsychiatry, University of Fukui, Eiheiji, Japan, **5** Center for Excellence in Thyroid Care, Kuma Hospital, Kobe, Japan

* kazsuzuki@shinshu-u.ac.jp

## Abstract

The COVID-19 pandemic significantly impacted psychological distress globally and led to widespread behavioral changes, including mask-wearing. Research shows that mask-wearing behavior may have psychological consequences. Infection control behaviors and psychological distress are expected to decrease as the pandemic subsides. However, the effect of these changes on patients with mental illnesses remains unclear. Hence, this study aimed to evaluate the impact of relaxed mask-wearing guidelines on psychological distress among psychiatric outpatients in Japan and its association with changes in mask-wearing behavior. It included 109 outpatients from a general hospital's psychiatric department. Psychological distress was assessed using the General Health Questionnaire-12 at two time points: before and after the guideline change. Mask-wearing behavior was recorded through self-reports. Changes in psychological distress were analyzed using the Wilcoxon signed-rank test, and the association between changes in mask-wearing behavior and psychological distress was examined using multiple regression analysis, adjusting for preceding psychological distress, age, and gender. Among the 109 participants (12 with schizophrenia, 55 with mood disorders, 34 with anxiety disorders, and 8 with other conditions), a significant reduction in psychological distress was observed after the guideline relaxation (Cohen's d = 0.344, p < 0.01). Outdoor mask-wearing decreased from 89% before the guideline change to 65% after the change. Changes in mask-wearing behavior were significantly associated with reduced psychological distress (β = 2.72, p < 0.01). Relaxed mask-wearing guidelines positively impacted psychological distress among psychiatric outpatients, with unmasking associated with improved mental health. Thus, the relaxation of public health measures can contribute to improved mental health among vulnerable populations. This study provides new insights into the psychological implications of mask-wearing policies in

**Data availability statement:** All relevant data are within the paper and its Supporting Information files.

**Funding:** The author(s) received no specific funding for this work.

**Competing interests:** The authors have declared that no competing interests exist.

the post-COVID-19 society and informs strategies to support mental health in future public health crises.

## Introduction

The COVID-19 pandemic, which began in 2019, has caused significant societal changes and adversely affected the mental health of individuals worldwide [1–4]. Several cohort studies initiated before the COVID-19 pandemic have shown that mental health problems either emerged or worsened during the pandemic [5,6]. Worsening psychological distress has been observed in several countries [7]. For example, in the United Kingdom, large population cohort data have shown increased psychological distress, as measured by General Health Questionnaire-12 (GHQ-12) scores during the pandemic [5]. Similarly, a Japanese cohort study reported worsening psychological distress evaluated by the Kessler Psychological Distress Scale after the COVID-19 pandemic [8]. Furthermore, longitudinal studies have identified a history of mental or physical health conditions and financial difficulties as contributing factors to persistent declines in mental health during the COVID-19 pandemic [9]. In the early stages of the COVID-19 pandemic, it was reported that environmental changes, such as isolation due to lockdown, exacerbated anxiety and depression symptoms, and a background of mental illness was an exacerbating factor [10]. These findings suggest that the pandemic's environmental changes had a particularly detrimental impact on the mental health of individuals who were already vulnerable, such as those affected by mental illnesses. Indeed, a Japanese study reported worsened psychological distress among individuals who visited a psychosomatic clinic for the first time after the pandemic compared to pre-pandemic levels [11]. The COVID-19 pandemic caused widespread fear and anxiety, leading to behavioral modifications, increased anxiety, social isolation, disrupted life rhythms, stigma, and social anxiety, resulting in changes in living conditions and increased psychological distress.

As an environmental change, the widespread practice of wearing masks during the COVID-19 pandemic may have had significant psychological repercussions. Wearing a mask has been shown to cause physical discomfort, such as difficulty breathing, ear pain, and headaches [12], which may influence emotional states and lead to psychological distress. At the same time, wearing a mask may provide a sense of protection, which can help reduce psychological stress [13]. During the COVID-19 pandemic, mask-wearing was recommended to prevent infection. People were expected to adjust their behavior depending on various conditions, including whether they lived in urban or rural areas, the infection status in their surroundings, and whether they were indoors or outdoors. Amid the intense media coverage, vulnerable individuals such as those with mental disorders may have experienced increased psychological stress due to heightened anxiety and greater sensitivity to how others perceived their mask-wearing behavior. In Japan, a high level of awareness of the need for masks existed as a cultural background even before the pandemic [14], and after the COVID-19 pandemic, the habit of wearing masks became even more deeply embedded in the population [15]. The common practice of wearing masks in

Japan has been considered to be linked to its collectivist culture [16]. Mask-wearing has a symbolic moral meaning, with some reports of its impact on psychological and behavioral aspects [17]. Since the beginning of the COVID-19 pandemic in Japan, even in less populated rural areas, people limited non-essential outings, took extra precautions while shopping, and regularly wore masks, establishing mask-wearing as a social norm across the country. These adaptive behaviors, represented by the wearing of masks encouraged during the COVID-19 pandemic, may have restricted people's freedom or induced psychological stress due to anxiety about peer pressure. Three years into the COVID-19 pandemic, as society was winding down from the COVID-19 turmoil, on March 13, 2023, the Japanese Ministry of Health, Labor, and Welfare announced that it would change its previous guidance strongly recommending the use of masks to respect individual choice and would weaken its stance of actively encouraging mask use [18]. Its mask use guidance clearly states that "mask use will largely depend on personal choice instead of current general requirements." It further limited the places where mask use is recommended to "medical institutions and nursing homes" and "in crowded trains and buses" only. However, in the months following the guideline change, it remains unclear whether the changed guideline actually changed mask-use behavior and whether it affected psychological distress of people with mental illnesses.

Here, we aimed to assess the impact of the relaxation of mask-use guidance by the Japanese government after the COVID-19 pandemic on the psychological distress of patients with mental illnesses. First, using the GHQ-12, we examined whether psychological distress differed before and after the changed mask-use guidance among psychiatric outpatients who visited a regional general hospital post-COVID-19 pandemic. Next, we examined whether psychological distress (measured by the GHQ-12) differed between patients who stopped wearing masks outdoors after the guideline change and those who continued to wear masks.

## Materials and methods

### Study setting and design

Participants were recruited from outpatient psychiatric patients attending Maruko Central Hospital, with the recruitment period starting on March 13, 2023, and ending on May 12, 2023. At the time of recruitment, GHQ-12 scores from the most recent visit were extracted from medical records as background data, and GHQ-12 scores after the guideline change were measured. A questionnaire regarding mask-wearing behavior before the guideline change was administered at the time of recruitment, and mask-wearing status was surveyed again at the outpatient visit three months after the guideline change. Maruko Central Hospital, the setting for this study, is the only general hospital in the Maruko region of Ueda City, Nagano Prefecture, which has a population of 20,191. Since Maruko Central Hospital does not have an inpatient psychiatric department, patients requiring psychiatric hospitalization are referred to nearby psychiatric institutions. Therefore, most of the participants in this study were homebound patients living in this rural area.

### Participants

Of the 139 patients who visited the psychiatric outpatient department of Maruko Central Hospital during the study period of two months after the relaxation of the mask-wearing guidance were announced, those with mental retardation (n = 5) and those with cognitive impairment (n = 7) were excluded due to concerns about the validity of the GHQ-12. Additionally, 18 patients with no GHQ-12 data prior to the guidance change were excluded. As a result, 109 patients were included in the final analysis (Fig 1). Diagnoses were made by psychiatrists based on the International Statistical Classification of Diseases and Related Health Problems, Tenth Edition (ICD-10). The study adhered to the ethical principles of the Declaration of Helsinki and was approved by the Ethics Committee of Maruko Central Hospital (approval numbers: 2022-Research 2 and 2023-Research 2). Informed consent was obtained in writing and explained verbally by the psychiatrist at the time of the visit. This consent procedure was also approved by the Ethics Committee of Maruko Central Hospital. Participation was voluntary, and data were treated anonymously.

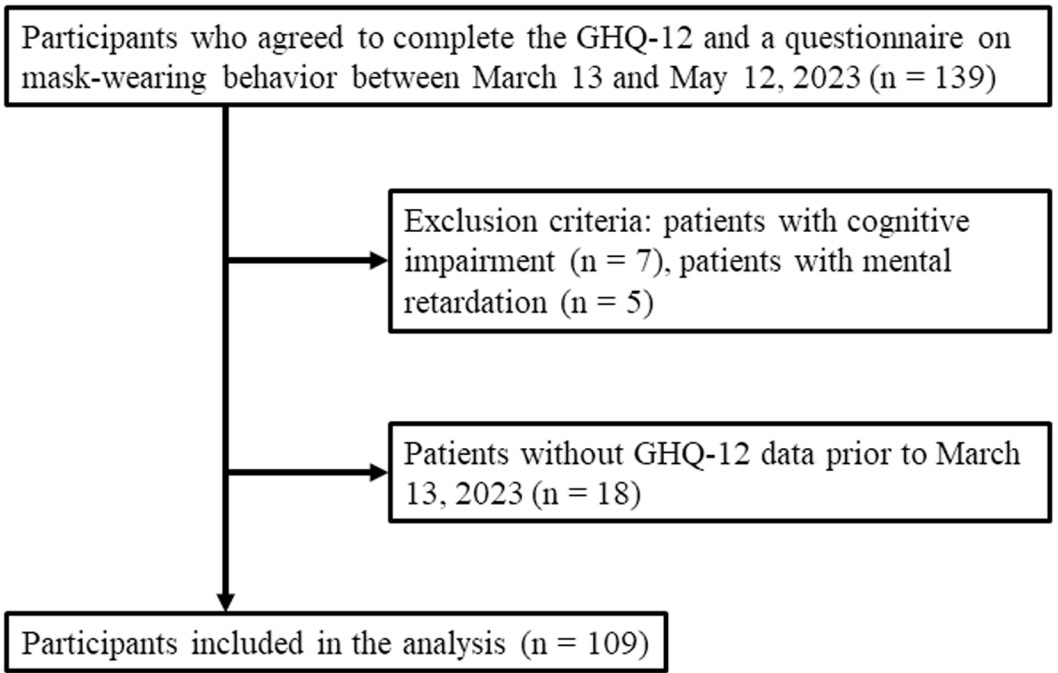

**Fig 1. Flow of selecting the study participants.** Note: Diagnoses were made by psychiatrists according to the ICD-10.

## Assessment of psychological distress

The GHQ-12 has shown its effectiveness across various population groups, cultures, and languages [19,20] as a simple, rapid and comprehensive screening tool for psychological distress and well-being. It has been used in studies targeting psychiatric patients with depression and anxiety disorders [21,22]. To assess psychological distress, we utilized the Japanese version of the GHQ-12 [23,24]. A previous study has confirmed its validity in the adult Japanese population [25]. The GHQ-12 items are scored on a scale as follows: 0 (not at all), 1 (no more than usual), 2 (rather more than usual), and 3 (considerably more than usual). We employed the GHQ scoring method to detect changes in mood, emotion, and behavior over the preceding four weeks, with a scoring range of 0-1-2-3. The total score was interpreted as an index of psychological distress.

## Assessment of mask-wearing behavior

During their visit to the psychiatric department from March 13 to May 12, the participants were asked to complete a questionnaire that asked whether they wore a mask at the hospital, at a shopping center, and outdoors before March 12. As a follow-up, they were asked to complete a questionnaire about their mask-wearing behavior on the closest outpatient visit three months later, after the guideline change. The responses were collected in a simple yes/no format regarding their mask-wearing behavior.

## Statistical analysis

Differences in psychological distress before and after the changed mask-wearing guideline were analyzed using the Wilcoxon signed-rank test since the assumption of normality was not fully met. Normality of GHQ-12 scores was assessed using the Shapiro–Wilk test, which showed no significant deviation at T1 (W = 0.979, p = 0.078) but a significant deviation at T2 (W = 0.966, p = 0.007). Skewness and kurtosis values also indicated a roughly normal distribution at T1

(skewness = 0.390, kurtosis = 0.017) and mild non-normality at T2 (skewness = 0.625, kurtosis = 0.798), further supporting the use of a non-parametric test for comparing T1 and T2 scores. To account for multiple hypothesis testing across the three situational conditions (outdoors, while shopping, and in hospitals), the Bonferroni correction was applied. The overall alpha level of 0.05 was divided by the number of comparisons, resulting in an adjusted threshold of $\alpha = 0.017$. Multiple regression analysis was conducted to examine the impact of changes in mask-wearing behavior on psychological distress, with two groups: those who continued to wear masks after the guidelines were issued and those who removed them when outdoors. The covariates included the initial GHQ-12 scores, gender, and age, with the GHQ-12 scores at post-guideline change as the outcome variable. The sample size calculation was performed using G*Power 3.1.9.7 [26,27]. Assuming a multiple regression model with explanatory variables for which we want to estimate the partial regression coefficients, and given an effect size of $f2 = 0.15$, when testing at a significance level of 5% and a power of 80%, the required sample size was calculated to be a total of 85 cases, indicating that the current sample size was sufficient for the analysis. The significance level ($\alpha$) was set at 0.05 for two-tailed tests. All statistical analyses were performed using the Statistical Package for the Social Sciences version 28.0 (SPSS 28.0) (IBM Corp., New York, USA). The graph in Fig 2 was created using GraphPad Prism 10 (GraphPad Software, San Diego, USA).

## Results

Of the participants, 44 were men and 65 were women, with an average age of 58.3 years. As shown in Table 1, the distribution of psychiatric diagnoses among the 109 psychiatric outpatients included schizophrenia (12 patients), mood

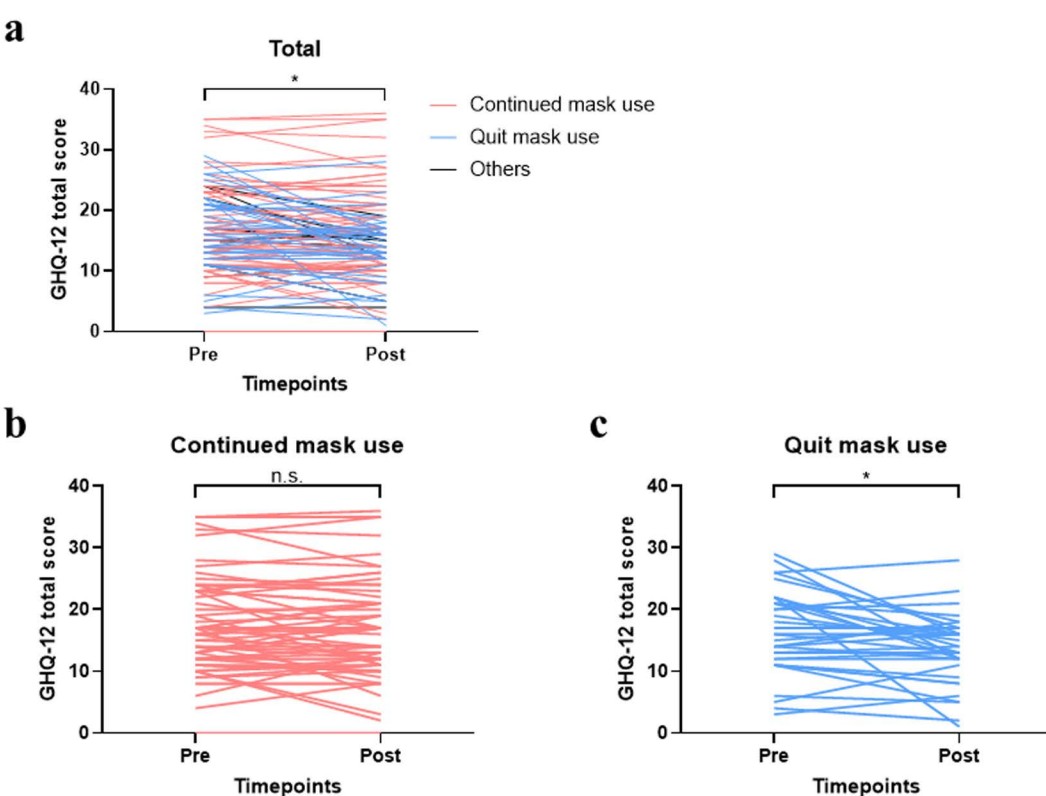

**Fig 2. Comparison of GHQ-12 total scores pre- and post- guideline change in the mask-wearing behavior.** "Others" refers to those who were not wearing masks before the relaxation of the mask-wearing guidelines. Note: $p < 0.01$, Wilcoxon signed rank test.

**Table 1. Characteristics of the participants.**

| Variable | | N | mean ± SD |
|---|---|---|---|
| GHQ-12 at pre-guideline change | | 109 | 16.7 ± 7.19 |
| GHQ-12 at post-guideline change | | 109 | 15.1 ± 7.12 |
| Age | | 109 | 58.3 ± 14.6 |
| Gender (men/women, N) | | 44/65 | |
| | | | wear/don't wear |
| Outdoor mask wearing behavior at pre-guideline change | | 109 | 97/12 |
| Outdoor mask wearing behavior at post-guideline change | | 109 | 65/44 |
| Diagnosis | Total | 109 | % |
| | Schizophrenia spectrum disorders[a] | 12 | 11 |
| | Mood disorders | 55 | 50.5 |
| | Neurotic, stress related and somatoform disorders | 34 | 31.2 |
| | Others | 8 | 7.3 |

SD: Standard deviation, GHQ-12: General Health Questionnaire-12

[a]Schizophrenia, schizotypal, and delusional disorders according to the ICD-10

disorders (55 patients), anxiety disorders (34 patients), and other conditions (8 patients). After applying the Bonferroni correction for multiple comparisons, a significant decrease in psychological distress, as assessed by the GHQ-12, was observed after the implementation of the government's mask-wearing mitigation guideline (Fig 2a). The mean GHQ-12 score decreased from pre- to post-guideline change, indicating a significant improvement in psychological distress (p < 0.01). Notably, the improvement in psychological distress was significant only in the group that stopped wearing masks outdoors (Figs 2b and 2c).

Regarding outdoor mask-wearing behavior at pre-guideline change, 97 of the 109 respondents reported wearing masks, while 12 reported not wearing them. At post-guideline change, 65 respondents reported wearing masks, while 44 reported not wearing masks. Furthermore, at pre-guideline change, 106 of the 109 respondents reported wearing masks when shopping indoors, indicating high compliance. Similarly, compliance with wearing masks at the hospital was also high, with 106 of 109 respondents reporting adherence. Hospitals requested patients to wear masks, and indeed, nearly all patients wore masks during outpatient visits.

Next, we hypothesized that behavioral changes associated with the relaxation of government mask use guidance might correlate with psychological distress. To verify this, we conducted multiple regression analysis using GHQ-12 at post-guideline change as the outcome for the 97 participants who wore masks at pre-guideline change. Sixty-two individuals who continued to wear masks outdoors from pre- to post-guideline change and 35 who stopped wearing masks outdoors were included in the analysis, and changes in mask-related behaviors remained significantly associated with psychological distress after adjustment for age, gender, and the GHQ-12 score at pre-guideline change (p < 0.01, Table 2). To assess whether the assumptions of the regression analysis were met, we conducted several diagnostic tests. Heteroscedasticity was examined by visually inspecting the residual plots, which showed no clear pattern suggesting non-constant variance (S1 Fig). The variance inflation factor (VIF) was also calculated for each independent variable. All VIF values were below 1.5, indicating no serious multicollinearity. In addition, we checked standardized residuals to identify potential outliers. No values exceeded ±3 standard deviations.

## Discussion

Psychological distress among psychiatric outpatients at Maruko Central Hospital improved following the relaxation of government mask use guidance on March 13, 2023. This suggests that the psychological distress of the study population,

**Table 2. Multiple linear regression analysis for psychological distress (GHQ-12 scores).**

| Variables | β | t | p | 95%CI | | | VIF |
|---|---|---|---|---|---|---|---|
| **Mask wearing behavior** | **2.717** | **2.892** | **0.005** | **0.851** | **–** | **4.584** | **1.047** |
| Age | −0.034 | −1.088 | 0.279 | −0.096 | – | 0.028 | 1.074 |
| Gender | 1.078 | 1.133 | 0.26 | −0.812 | – | 2.967 | 1.098 |
| GHQ-12 at pre-guideline change | 0.779 | 12.701 | <0.001 | 0.657 | – | 0.901 | 1.047 |

Comparison of those who continued to wear masks and those who stopped wearing it

GHQ-12: General Health Questionnaire-12, CI: confidence interval, VIF: variance inflation factor

outpatients with psychiatric disorders, may have been positively affected by the relaxation of the behavioral guidelines. Considering the numerous studies showing worsening psychological distress before and during the COVID-19 pandemic [5–7,28], the results of this study highlight the close relationship between changes in social behavior and psychological problems. Additionally, multiple regression analysis of the mask-wearing group at pre-guideline change showed that changes in mask-wearing behavior were associated with psychological distress at post-guideline change, and that this longitudinal association remained after adjusting for psychological distress at pre-guideline change. In other words, among those who continued to wear masks during the COVID-19 pandemic, stopping mask use may have led to a reduction in psychological distress. To our knowledge, this study is the first to show that post-pandemic reintegration efforts improved psychological distress among outpatient with mental illnesses in Japan.

There may be direct and indirect effects that contribute to the association between mask-wearing behavior and improved psychological distress. One possibility is that relaxed mask-wearing guidelines directly improved psychological distress by reducing the extra effort and discomfort for those unaccustomed to wearing masks [29]. Another indirect effect is that some people who failed to cope with or over-adapted to the initial behavioral changes of the COVID-19 pandemic may have experienced psychological distress, and the relaxation of mask-wearing guidelines may have alleviated these maladjustments. In other words, the guidelines on mask-wearing behavior may have reduced psychological distress by providing a valid reason to stop wearing masks for individuals who were hesitant to wear masks but were forced to do so by social peer pressure since the early period of the COVID-19 pandemic. Moreover, decreased anxiety about unfamiliar viruses and reduced awareness of social surveillance due to revised guidelines may have positively affected the levels of psychological distress. In addition, other intraindividual factors, such as each person's level of anxiety, depression, and obsession, may have also played a role in GHQ-12 scores and behavioral changes. Genetic background may also be involved as one of the intraindividual factors. In fact, it has been reported that the Japanese have a higher frequency of the short allele of the 5-HTTLPR gene, which is associated with anxiety and collectivistic cultural values [30]. Since the present data were obtained from patients with psychiatric disorders in a single region in Japan, there may be a stronger genetic background. Because the present study did not directly measure anxiety, depression, obsessions, or genetic background as described above, the preceding GHQ-12 score was used as a covariate to control for individual factors. Still, the results remained significant. Altogether, government changes in mask use guidance may have altered the mask-wearing behavior among outpatients with mental illnesses and thus ameliorated psychological distress.

Even before the COVID-19 pandemic, mask-wearing in Japan was reported as a "unique and enigmatic social norm" [14]. The Japanese practice of wearing masks has been attributed to a collectivistic culture that prioritizes conformity and harmony within groups over individualism [16]. Because of this cultural background, the wearing of masks was immediately encouraged during the COVID-19 pandemic, with most people complying due to an implicit social understanding [15]. Somewhat excessive reactions also occurred, causing social confusion, such as the panic buying of masks due to anxiety about a shortage, and leading to a shortage of masks even in hospitals [31]. The present study surveyed outpatients with psychiatric disorders in a rural area of Japan and found that 88% wore masks outdoors and 98% wore masks

while shopping during the COVID-19 pandemic. Although this study was conducted in a rural area within Japan, a nation-wide internet-based report from the same period in February 2023 indicated that 4.9% of people did not wear masks at all during a specific 9-day period [32], making the present survey comparable to this report in terms of the high wearing rate. Furthermore, the highest mask-wearing rates reported in other countries were 77% in February 2021 in the U.S. and 86% in December 2020 in Canada, suggesting that Japan had particularly high mask-wearing rates [33]. In this cultural context, government encouragement of mask-wearing behavior may have led to changes in mask-wearing behavior, which in turn may have alleviated the fear of infection.

Government intervention against COVID-19 infection in the early stages of the pandemic has been reported to improve mental health [34]. The prevalence of mask-wearing behavior has been reported to vary by local culture [35], and educational interventions on mask-wearing in rural areas have been observed to increase mask-wearing rates and decrease the risk of COVID-19 infection [36]. Wearing masks is a simple and effective way to prevent COVID-19 infection. Some countries issued guidelines recommending the use of masks immediately after the pandemic onset, while others did not, resulting in varied mask use measures depending on infection status and time factors [37,38]. The World Health Organization was initially cautious in recommending the use of masks but updated its guidelines on June 5, 2020, to recommend mask use when it is difficult to maintain physical distance [39]. When the pandemic began to subside, responses varied, with governments differing on the appropriateness and timing of lifting mandatory mask use in public places. While some countries immediately lifted the requirement [40], the Japanese government was relatively cautious in its response. Finally, three years after the onset of the COVID-19 pandemic, on March 13, 2023, the Japanese Ministry of Health, Labor, and Welfare announced that the government was weakening its active encouragement of mask use by issuing guidelines that emphasized individual choice in mask use [18]. In our study population, outdoor mask use decreased from 89% to 59% following the government guidelines. A previous study in Japan using data from before and after the implementation of these guidelines has suggested a bidirectional relationship between mask-wearing and the reasons for wearing masks [41]. Taken together, providing such guidelines and information on how to downsize mask-wearing at the appropriate time to reduce social tension and improve well-being due to the COVID-19 pandemic should be a step forward toward social normalization. This government's initiative to provide concrete guidelines for action may have provided some relief to people with mental illness. The results of this study suggest that changes in policy guidelines may also affect the mental health of vulnerable individuals residing in rural areas. Policies are typically formulated with higher consideration of urban residents, but the impact on vulnerable groups in rural areas—who often have limited access to information and tend to hold conservative attitudes—should not be overlooked. Furthermore, healthcare professionals working in mental health care in these regions must understand the psychological care needs of these individuals, which is essential for providing appropriate psychological care to those with mental illnesses.

## Limitation

This study has several limitations that deserve consideration. First, the study was conducted at a single center in rural Japan, which may introduce selection bias and limit the generalizability of the findings to other regions or populations. However, mask-wearing rates were found to be consistent nationwide, and the homogeneity of the study population may also serve as a strength in controlling for regional variability. Second, the assessment of mask-wearing behavior relied on self-reported data, which may be subject to recall bias. Although participants were asked about recent and clearly defined behaviors, some degree of misreporting cannot be ruled out. Third, unmeasured confounding factors such as ongoing treatment effects, changes in medication, seasonal influences, and external events (e.g., national celebrations, including Japan's victory in the World Baseball Classic or other social events) may have impacted the outcomes. While we included baseline GHQ-12 scores as a covariate to partially account for individual differences in psychological distress, residual confounding may still exist. Lastly, the relatively small

sample size limited our ability to conduct subgroup analyses by diagnosis or to examine long-term effects. Future multi-center studies with larger and more diverse populations, as well as longitudinal designs, are needed to confirm and expand upon these findings.

## Conclusions

Given the psychological consequences of environmental behavioral changes due to the COVID-19 pandemic among patients with mental illnesses, this study aimed to evaluate the impact of relaxed mask-wearing guidelines on psychological distress among psychiatric outpatients in Japan and its association with changes in mask-wearing behavior. Our findings indicate a significant reduction in psychological distress and a decrease in outdoor mask wearing after the guideline relaxation. Moreover, changes in mask-wearing behavior were significantly associated with reduced psychological distress, while relaxing of mask-wearing guidelines positively impacted psychological distress among psychiatric outpatients, with unmasking associated with improved mental health. Our results confirm that the relaxation of public health measures can contribute to improved mental health in vulnerable populations. Hence, we recommend administrations and governments provide appropriate guidelines that align with the local cultural context in the event of health disasters, with a focus on improving mental health and well-being.

## Supporting information

**S1 Table. Analysis sample data.** Total sample data used for analysis. Note: '0' indicates wearing a mask, and '1' indicates not wearing a mask.
(XLSX)

**S1 Fig. P–P and Q–Q plots for assessing normality of residuals.**
(TIF)

## Acknowledgments

We express our sincere thanks for the technical and clinical assistance provided by Misawa, Shimizu, and Horiuchi.

## Author contributions

**Conceptualization:** Kazuhiro Suzuki, Yuka Mizuno.

**Data curation:** Kazuhiro Suzuki, Yuka Mizuno.

**Formal analysis:** Kazuhiro Suzuki, Yusuke Arai, Keitaro Miyamura.

**Investigation:** Kazuhiro Suzuki, Yuka Mizuno.

**Methodology:** Kazuhiro Suzuki, Yusuke Arai, Keitaro Miyamura, Daimei Sasayama, Shinsuke Washizuka.

**Project administration:** Kazuhiro Suzuki.

**Resources:** Kazuhiro Suzuki.

**Software:** Kazuhiro Suzuki.

**Supervision:** Yusuke Arai, Daimei Sasayama, Shinsuke Washizuka.

**Validation:** Keitaro Miyamura.

**Visualization:** Kazuhiro Suzuki.

**Writing – original draft:** Kazuhiro Suzuki.

**Writing – review & editing:** Yusuke Arai, Keitaro Miyamura, Daimei Sasayama, Shinsuke Washizuka.

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
