## [Decision Letter · Decision Letter 0]

21 May 2025

PONE-D-24-41871Longitudinal association between psychological distress and mask-wearing post COVID-19 among psychiatric outpatients in JapanPLOS ONE

Dear Dr. Suzuki,

Thank you for submitting your manuscript to PLOS ONE. After careful consideration, we feel that it has merit but does not fully meet PLOS ONE’s publication criteria as it currently stands. Therefore, we invite you to submit a revised version of the manuscript that addresses the points raised during the review process.

We look forward to receiving your revised manuscript.

Kind regards,

Bimala Panthee

Academic Editor

PLOS ONE

Journal Requirements:When submitting your revision, we need you to address these additional requirements. 1. Please ensure that your manuscript meets PLOS ONE's style requirements, including those for file naming. The PLOS ONE style templates can be found at  https://journals.plos.org/plosone/s/file?id=wjVg/PLOSOne_formatting_sample_main_body.pdf and https://journals.plos.org/plosone/s/file?id=ba62/PLOSOne_formatting_sample_title_authors_affiliations.pdf 2. We suggest you thoroughly copyedit your manuscript for language usage, spelling, and grammar. If you do not know anyone who can help you do this, you may wish to consider employing a professional scientific editing service.  The American Journal Experts (AJE) (https://www.aje.com/) is one such service that has extensive experience helping authors meet PLOS guidelines and can provide language editing, translation, manuscript formatting, and figure formatting to ensure your manuscript meets our submission guidelines. Please note that having the manuscript copyedited by AJE or any other editing services does not guarantee selection for peer review or acceptance for publication.  Upon resubmission, please provide the following: • The name of the colleague or the details of the professional service that edited your manuscript• A copy of your manuscript showing your changes by either highlighting them or using track changes (uploaded as a *supporting information* file)• A clean copy of the edited manuscript (uploaded as the new *manuscript* file) 3. We note that there is identifying data in the Supporting Information file < S1table.xlsx>. Due to the inclusion of these potentially identifying data, we have removed this file from your file inventory. Prior to sharing human research participant data, authors should consult with an ethics committee to ensure data are shared in accordance with participant consent and all applicable local laws. Data sharing should never compromise participant privacy. It is therefore not appropriate to publicly share personally identifiable data on human research participants. The following are examples of data that should not be shared: -Name, initials, physical address-Ages more specific than whole numbers-Internet protocol (IP) address-Specific dates (birth dates, death dates, examination dates, etc.)-Contact information such as phone number or email address-Location data-ID numbers that seem specific (long numbers, include initials, titled “Hospital ID”) rather than random (small numbers in numerical order) Data that are not directly identifying may also be inappropriate to share, as in combination they can become identifying. For example, data collected from a small group of participants, vulnerable populations, or private groups should not be shared if they involve indirect identifiers (such as sex, ethnicity, location, etc.) that may risk the identification of study participants. Additional guidance on preparing raw data for publication can be found in our Data Policy (https://journals.plos.org/plosone/s/data-availability#loc-human-research-participant-data-and-other-sensitive-data) and in the following article: http://www.bmj.com/content/340/bmj.c181.long. Please remove or anonymize all personal information (Age), ensure that the data shared are in accordance with participant consent, and re-upload a fully anonymized data set. Please note that spreadsheet columns with personal information must be removed and not hidden as all hidden columns will appear in the published file.

**Additional Editor Comments:**

Please  incorporate all the comments raised by our reviewers. Further, justification of the study in introduction section needs more elaboration and validity and reliability of the study measures needs to be mentioned .Also mention about the management of  biases  in this study that might have impacted in the study findings.. 

Reviewers' comments:

Reviewer's Responses to Questions

**Comments to the Author**

1. Is the manuscript technically sound, and do the data support the conclusions?

Reviewer #1: Yes

Reviewer #2: Yes

2. Has the statistical analysis been performed appropriately and rigorously? 

Reviewer #1: Yes

Reviewer #2: Yes

3. Have the authors made all data underlying the findings in their manuscript fully available?

Reviewer #1: No

Reviewer #2: Yes

4. Is the manuscript presented in an intelligible fashion and written in standard English?

Reviewer #1: Yes

Reviewer #2: Yes

5. Review Comments to the Author

Reviewer #1: The Shapiro-Wilk test checks if the data are normally distributed, but it doesn't mention other important values, such as skewness or kurtosis. Non-parametric tests (Wilcoxon) solve this problem, but it would be better to include these indicators.

Residual plots or tests for heteroscedasticity, multicollinearity (VIFs), or outliers were not discussed. These are critical for validating regression assumptions.

The study examined multiple outcomes (mask wear in different settings: outdoors, shopping, hospitals). If you run multiple tests without correcting for them (like using Bonferroni correction), there is a chance that the number of false positives ( Type I errors) will increase.

Subgroup analyses by diagnosis (e.g., mood vs. anxiety disorders) were not performed. It may be due to sample size constraints. So you can include it as recommendation for future studies could explore these.

So,

Clarify normality test results or justify non-parametric choices more explicitly.

Report model diagnostics (e.g., residuals, VIFs) for regression.

Address potential multiple testing issues if secondary analyses were exploratory.

Reviewer #2: Introduction: Your study raises an important discussion about the potential distress associated with mask usage in vulnerable populations. However, one key area that could benefit from further elaboration is the justification for why mask-wearing leads to psychological distress specifically among psychiatric patients.

Line 102-106 appears misplaced.

Results: Line198-200 appears repetitive (data provided in the table)

Discussion: Your findings highlight an important issue, but I believe the discussion could benefit from a stronger emphasis on the clinical and practical implications of your study for psychiatric care.

Currently, while the results are well presented, there is limited exploration of how these findings could inform mental health interventions, policy adjustments, or clinical practices.

6. PLOS authors have the option to publish the peer review history of their article (what does this mean? ). If published, this will include your full peer review and any attached files.

**Do you want your identity to be public for this peer review?** For information about this choice, including consent withdrawal, please see our Privacy Policy .

Reviewer #1: No

Reviewer #2: No

---

## [Author Response · Author response to Decision Letter 1]

4 Jul 2025

We sincerely thank the editors and reviewers for their valuable and constructive feedback. To enhance the clarity and readability of the manuscript, we employed a professional English editing service (Editage) and made several minor revisions. In response to concerns regarding the anonymization of Table S1, we have removed all personally identifiable information to ensure full compliance with data privacy requirements.

Detailed responses to all reviewer and editor comments are provided in the rebuttal letter. Please do not hesitate to contact us should you have any further feedback.

---

## [Editor Report · Decision Letter 1]

20 Jul 2025

Longitudinal association between psychological distress and mask-wearing post COVID-19 among psychiatric outpatients in Japan

PONE-D-24-41871R1

Dear Dr. Suzuki,

We’re pleased to inform you that your manuscript has been judged scientifically suitable for publication and will be formally accepted for publication once it meets all outstanding technical requirements.

Kind regards,

Bimala Panthee

Academic Editor

PLOS ONE
---

## [Editor Report · Acceptance letter]

PONE-D-24-41871R1

PLOS ONE

Dear Dr. Suzuki,

I'm pleased to inform you that your manuscript has been deemed suitable for publication in PLOS ONE. Congratulations! Your manuscript is now being handed over to our production team.

Kind regards,

on behalf of

Dr. Bimala Panthee

Academic Editor

PLOS ONE